# Design and Synthesis of C-O Grain Boundary Strengthening of Al Composites

**DOI:** 10.3390/nano10030438

**Published:** 2020-02-29

**Authors:** Jianian Hu, Jian Zhang, Guoqiang Luo, Yi Sun, Qiang Shen, Lianmeng Zhang

**Affiliations:** State Key Laboratory of Advanced Technology for Materials Synthesis and Processing, Wuhan University of Technology, Wuhan 430070, China; hujianian1@outlook.com (J.H.); Zhangjian178@whut.edu.cn (J.Z.); sunyiwhut@163.com (Y.S.); sqqf@263.net (Q.S.); lmzhang@whut.edu.cn (L.Z.)

**Keywords:** Al matrix, C-O strengthening design, reaction simulation, Al_4_C_3_, Al-O solution, strengthening mechanism

## Abstract

This research presents an approach for C-O grain boundary strengthening of Al composites that used an in situ method to synthesize a C-O shell on Al powder particles in a vertical tube furnace. The C-O reinforced Al matrix composites (C-O/Al composites) were fabricated by a new powder metallurgy (PM) method associated with the hot pressing technique. The data indicates that Al_4_C_3_ was distributed within the Al matrix and an O-Al solution was distributed in the grain boundaries in the strengthened structure. The formation mechanism of this structure was explained by a combination of TEM observations and molecular dynamic simulation results. The yield strength and ultimate tensile strength of the C-O/Al composites, modified by 3 wt.% polyvinyl butyral, reached 232.2 MPa and 304.82 MPa, respectively; compared to the yield strength and ultimate tensile strength of the pure aluminum processed under the same conditions, there was an increase of 124% and 99.3%, respectively. These results indicate the excellent properties of the C-O/Al-strengthened structure. In addition, the strengthening mechanism was explained by the Hall–Petch strengthening, dislocation strengthening, and solid solution strengthening mechanisms, which represented contributions of nearly 44.9%, 15.9%, and 16.6% to the total increased strength, respectively. The remaining increment was attributed to the coupled strengthening of the C and O, which contributed 20.6% to the total increase.

## 1. Introduction

Grain boundaries are defective in most engineering materials and control numerous mechanical, functional, and dynamic properties [1,2]. The performance of materials can be influenced by local chemical elements, such as B [3,4], C [5,6,7], N [8,9], and O [10,11,12] nonmetallic impurities, which are introduced into the grain boundaries by conventional processing methods and working environments. For example, steels can substantially increase their toughness and hardness by introducing suitable amounts of C and B elements [13]. In addition, in the presence of process, control agents and impurities (such as C, O, and H) are usually incorporated into materials produced by mechanical alloying and introducing nonmetallic compounds [14,15,16]. In addition to improving the mechanical properties, the introduction of impurities can increase GB cohesion, significantly limit grain growth, and increase the thermal stability [17,18,19]. Therefore, it is very important to design and optimize elemental segregation at the grain boundaries to improve the material properties.

The literature results show that the distribution of C and O elements at the grain boundaries of the Al phase can significantly decrease the stacking fault energy [20]. In addition, O at the grain boundaries can significantly enhance the shear strength of the Al matrix by the solution strengthening mechanism. The C element at the grain boundaries reacts with Al to generate Al_4_C_3_, which enhances the strength of the Al matrix by precipitation strengthening and dislocation strengthening [11,12,21,22]. Therefore, it is reasonable to design a C-O coupling enhancement to the Al matrix with a specific microstructure, as shown in Figure 1a. The goal is to produce a specific strengthened microstructure comprising an Al-O solution and nanoparticles of Al_4_C_3_.

According to recent reports [23,24,25,26], C/Al composites can be produced by directly introducing different carbon structures (such as carbon nanotubes, graphite powders, and activated carbon flakes) into aluminum with powder metallurgy fabrication methods; however, it is difficult to form a strong interface between aluminum and carbon upon mixing due to the low wettability between aluminum and carbon. Thus, O is often introduced to an Al matrix by oxidation in an ambient atmosphere, but it is difficult to control the amount of O upon oxidation, and the O often reacts with Al to form nanoparticles of Al_2_O_3_ [12,27]. To overcome the problems of wettability between aluminum and carbon and control the amount of O that is incorporated, an in situ reaction was designed to produce a specific strengthened microstructure comprising an Al-O solution and nanoparticles of Al_4_C_3_. The reaction included two steps as follows:(1)Al + O → AlO(solution)
(2)$$\mathrm{Al}\;+\;C\;\to \;{\mathrm{Al}}_{4}C_{3}$$

Introducing polymers onto Al particles is a very effective way to easily achieve doping of trace elements on metal surfaces. Fluoropolymers have improved the oxidation efficiency of Al particles by introducing trace nonmetallic elements that contain C, O, and F on the surface of the Al particles [28,29]. Part of the C, H, and O was effectively removed through degradation, and the species that remained were primarily C-O-F. Thus, it is reasonable to design a C-O nanoshell on Al particles by introducing polymers onto the Al particles, as shown in Figure 1b. In addition, the reaction between the Al and C-O shell should also be controlled, and reaction (1) should first be activated; otherwise, the reaction product of Al_4_C_3_ prevents the O from being incorporated into the Al matrix. Therefore, the coating polymers should be deoxygenated easily. PVB is a very common source of C and can introduce a small amount of O. Therefore, PVB is often used to coat other metals to provide a straightforward carbon source. The O element on PVB polymers is unstable and easily activates deoxygenation. The reaction process was illustrated by molecular dynamic simulation, and the simulation results suggested that reaction (1) was first activated, as shown in Figure 1c.

In our research, PVB was used to coat Al particles in a solvent, and the PVB coating on the Al particles was vacuum dried and degraded. A C-O shell was generated on the Al particles, and the modified Al particles were consolidated by hot pressing. The specific sintering mechanism was indicated by the HRTEM results and molecular dynamics (MD) simulation.

## 2. Materials and Methods

The raw aluminum powder (average particle size was approximately 2–3 μm, purity >99.9%, Bai Nian Ying, Zhejiang, China) was directly mixed with different polyvinyl butyral contents (1.5 wt.%, 3 wt.%, and 4.5 wt.%, Aladdin, Shanghai, China) in ethanol. The experimental procedure is shown in the schematic diagram in Figure 1. The mixture was dried in an oven at 80 °C for 8 h to ensure complete evaporation of the ethanol solvent. Then, the aluminum powder and polyvinyl butyral mixture was placed in a furnace at 480 °C for 1 h in a vacuum atmosphere to transform the organic additives into a C-O coating. The C and O content of the modified powders were performed on a CHONS analyzer (Vario EL cube, Germany). The quantity of the nanoparticles distribution of Al4C3 in the grain as well as the Al-O-Al distribution can be evaluated by the C and O content, supposing the reaction had gone completely. Cand O content of the modified powder and calculated quantity of the Al-O solution and Al_4_C_3_ in the composites is listed in Table 1. Then, the C-O coated aluminum powder was placed into a tungsten carbide (WC) dye with a diameter of 10 mm under a pressure of 30 MPa; the green compact was a cylinder with a bottom diameter of 10 mm. Finally, the compact was heated in a vacuum hot-pressing furnace at a rate of 10 °C/min and sintered at 630 °C. The microstructure of the interface was observed on a TEM machine (JEM-2100F STEM, Japan). The tensile strength of the specimens was on a universal test system (Instron-5966, Boston, MA, USA).

## 3. Results and Discussion

A molecular dynamics (MD) simulation of the reaction between the vinyl butyral monomer (the monomer of PVB) and Al was performed to understand our experimental observations, such as the O absorption mechanism and carbon chain aggregation. The MD simulations used a reactive force field (ReaxFF) potential for the calculations. The chemical reactions in the Al-C-H-O system can be accurately described with the ReaxFF atomistic potentials [30]. The model comprised a vinyl butyral monomer above an Al plane, as shown in Figure 2a. The system was heated to 800 K in an NVE Ensemble system. From the MD result of the reactions of the Al-C-H-O system, the O dissociation on the vinyl butyral monomer was first activated by forming two C chains, illustrated in Figure 2b,e. Then, the O was absorbed by the Al to form an Al-O solution, and the C chains dissociated and were kept away from the Al particle surface, which is shown in Figure 2c,d. The reactions can be described as follows:
(3)$$C_{6}H_{16}O_{2}\;\to \;O\cdot \cdot +{\;C}_{4}H_{8}\cdot \cdot$$
(4)Al + O → AlO(solution)

The reaction suggests that PVB is an ideal polymer to produce the final strengthening microstructure.

PVB was therefore used to modify the Al particles. The TEM images show the PVB-coated Al particles after degradation; the C-O coated Al samples and EDS results can be seen in Figure 3. The results show that the particles were spherical and monodispersed with a particle size of 2–3 μm. Due to the aggregation of the PVB during the mixing process, there were carbon-rich locations in the coating. In the case of the Al–C sample, each of the particles consisted of an Al core, carbon shell, and C-O shell coating with a thickness of approximately 10 nm, as shown in Figure 3b. The EDS spectra in Figure 3c,d show direct evidence of the existence of Al and C. In addition, the C signal can be observed in the EDS spectra in the locations that contained aggregations, further confirming that the aggregated phase was rich in carbon. The amount of O was too low to be detected in the EDS spectra.

Figure 4 shows the transmission electron microscopy topography of the sintered sample modified by 3 wt.% PVB at 630 °C. From the TEM image and EDS results of the composite in Figure 4a–c, the Al matrix was surrounded by O at the grain boundaries. To obtain a detailed understanding of the O structure, HRTEM was used to study the structure of the grain boundaries, as shown in Figure 4d. A nanoparticle with a size of 3 nm was observed in a grain boundary of the Al matrix, and the FTT electron diffraction pattern of the nanoparticle was that of Al, suggesting that the nanoparticle comprised a solid solution of Al. The C signal was detected in the TEM image in Figure 4e, the Al/Al grain boundaries were very pure, and the only interface phase that was present was a rod-like new phase. This phase was determined to be the Al_4_C_3_ phase from the electron diffraction results, and the lattice spacing was found to be 0.833 nm, which corresponds to the (003) plane of the Al_4_C_3_ crystal.

By combining the experimental results and MD simulation results, the formation mechanism of the microstructure observed herein can be explained in detail. The PVB polymer coated the surface of the Al particles and formed a core shell model. With increasing temperature, O dissociation on the vinyl butyral monomer was first activated by the formation of two C chains, and then the O was mostly absorbed into the Al crystal grain boundaries, which is shown in Figure 4d. The literature indicates that the stability of this structure is high and that O atoms did not migrate into the interior of the crystal readily [11]. Therefore, the carbon content of the external polymer residue was high. Due to the ability of transferring lone electron pairs from O to Al, the oxygen was electronegative δ^-^, and the aluminum was electropositive δ^+^. The O acted as a link between the Al particles and the external polymer, and the reduction of the O in the external polymer caused the affinity of the outer shell layer to be further reduced. Therefore, with increasing temperature, the oxygen content of the shell layer was reduced, the wettability of the outer polymer shell was gradually reduced, and the external carbon chains tended to self-aggregate, thus reacting with Al to form Al_4_C_3_ in a dispersed state.

The tensile properties of the fabricated different C-O/Al composites modified by different PVB content and reference Al materials are shown in Figure 5. From the tensile stress–strain curves of the fabricated materials in Figure 5, the yield strength (YS) and ultimate tensile strength (UTS) of the C-O/Al composites modified by 3 wt.% PVB were 228.9 MPa and 304.8 MPa, respectively. In addition, compared with the YS and UTS of the pure aluminum processed under the same conditions, 102.1 MPa and 152.9 MPa, respectively, there was an increase of 124% and 99.3%, respectively. The C-O/Al composite modified by 4.5 wt.% PVB was turned out to have a stronger YS of 242.5 MPa and a lower UTS of 260.1 MPa than the C-O/Al composite modified by 3 wt.% PVB, which was explained by brittle phases aggradation on the grain boundaries. The tensile properties of the C-O/Al composite modified by 3 wt.% PVB content were considered to be relatively high when the designed phase was in a dispersed distribution.

In the C-O/Al composite modified by 3 wt.% PVB, different mechanisms for enhancing the grain boundaries were activated to obtain the strengthening effect, including Hall–Petch strengthening Δσ_HP_, dislocation strengthening Δσ_d_, and solid solution strengthening Δσ_ss_. The contributions of each mechanism to the reinforcement of the system were quantified in the present study by using a unique methodology.

The initial C-O species at the grain boundaries limited the growth of the aluminum particles. The strengthening effect is illustrated by the relation of Hall-Petch strengthening, which is shown in Equation (3) [31]:(5)$$∆\sigma_{\mathrm{HP}}=\frac{k_{y}}{\sqrt{d_{1}}}-\frac{k_{y}}{\sqrt{d_{0}}}$$
where d_1_ is the average grain size of the Al matrix after the C-O strengthening and d_0_ is the average grain size of the Al matrix without the presence of PVB. For the Al alloy, k_y_ is the Hall–Petch coefficient and was equal to 0.22 MPa [32]. The value for the Hall–Petch strengthening, Δσ_HP_, was calculated to be 57 MPa.

For an alternative way to enhance metals, we assumed a classic distortion interaction with a solution atmosphere, and the strengthening resulting from the presence of solutes is often generally expressed as Equation (4) [11]:(6)$$∆\sigma_{\mathrm{ss}}={\;\beta G\varepsilon }^{p}c^{q}$$
where G is the shear modulus of the solvent, which is 26.9 GPa for Al series alloys; c is the concentration of the solute; β is an empirically determined proportionality constant related to the obstacle strength, which is often equal to 0.1; and ε is referred to as an interaction parameter. The parameter ε reduces to the misfit strain quantity $\varepsilon =\left|\left(r_{\mathrm{matrix}}-{\;r}_{\mathrm{solution}}\right)\;/{\;r}_{\mathrm{matrix}}\right|$, where r_matrix_ and r_solution_ are the atomic radii of the matrix and solution, respectively. Parameters p and q are constants that describe the solute spacing and dislocation solute statistical mean values, respectively. The Fleischer model was used to define the constant in the model, and p = 3/2 and q = 1/2 [33]. Δσ_ss_ was calculated to be 20.19 MPa.

The parameter Δσ_d_ is the strength increment due to the CTE mismatch between the matrix and the reinforcement upon cooling. Considering the dislocation enhancement discussed above, the dislocation strengthening model is given by Equation (5) [34]:(7)$$∆\sigma_{d}=\;\mu b\sqrt{\rho }$$
where η is a geometric constant and is equal to 1.25 for a fcc metal, μ is the shear modulus of the matrix and is equal to 2.64 × 10^10^ N/m^2^, b is the magnitude of the Burgers vector and is equal to 0.286 nm for Al, and ρ is the dislocation density [29].

Furthermore, considering the thermal mismatch at the interface of the Al_4_C_3_ reinforcements and Al matrix, the dislocation density can be calculated by Equation (6) [35]:(8)$$\rho \;=\frac{4V∆\alpha ∆T}{b\left(1-V\right)}(\frac{1}{t_{1}}+\frac{1}{t_{2}}+\frac{1}{t_{3}})$$
where V is the volume fraction of the reinforcement, Δα is the difference between the thermal expansion coefficient of the matrix and reinforcement, ΔT is the difference between the sintering temperature and room temperature, b is the Burgers vector of the Al matrix, t represents the dimension of the reinforcement (t_1_ is the length of the reinforcement particle, t_2_ is the width, and t_3_ is the height), and Δσ_d_ was calculated to be 21.08 MPa.

In conclusion, the greatly increased tensile strength of C-O/Al composites was due to Hall–Petch strengthening Δσ_HP_, dislocation strengthening Δσ_d_, and solid solution strengthening Δσ_ss_, and each contributed 44.9%, 15.9%, and 16.6%, respectively. The remaining increment may be attributed to strengthening from the C and O, which contributed 20.6% to the total strength.

## 4. Conclusions

The YS and UTS of the Al composites modified by 3 wt.% PVB were 228.9 MPa and 304.82 MPa, respectively; compared to the YS and UTS of pure aluminum processed under the same conditions, they experienced an increase of 124% and 99.3%, respectively. This result indicates that the C-O/Al structure provided strengthening.

The results from the analysis of the strengthened structure suggests that Al_4_C_3_ was distributed within the Al matrix with an Al-O solution present around the grain boundaries.

The formation mechanism of this structure was explained by a combination of TEM observations and molecular dynamic simulation results. The Al-O solution reaction was first activated; then, the remaining C tended to aggregate and react with Al, generating Al_4_C_3_ at the grain boundaries.

The strengthening mechanism of the microstructure was explained by the Hall–Petch, dislocation strengthening and solid solution strengthening mechanisms, which contributed 44.9%, 15.9%, and 16.6% of the total strengthening, respectively.

## Figures and Tables

**Figure 1 nanomaterials-10-00438-f001:**
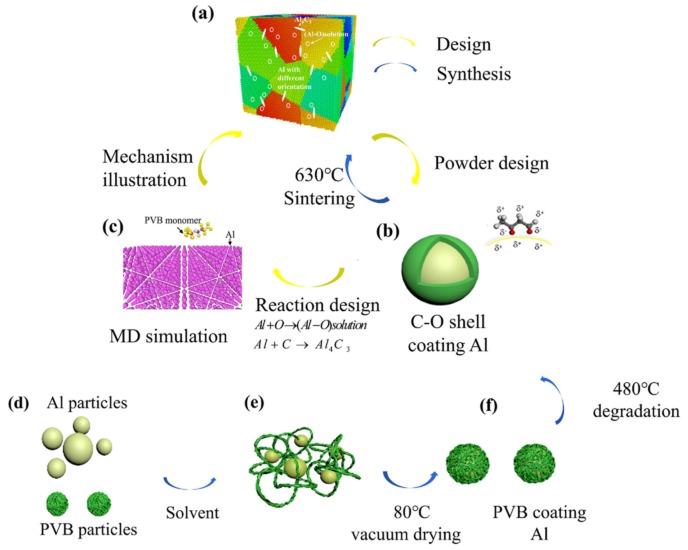
Schematic of the structural design and powder preparation flow chart: (**a**) designed structure of the C-O/Al composite, (**b**) designed powder structure, (**c**) molecular simulation of the reaction design, (**d**), (**e**), and (**f**) schematic diagram of the powder pretreatment.

**Figure 2 nanomaterials-10-00438-f002:**
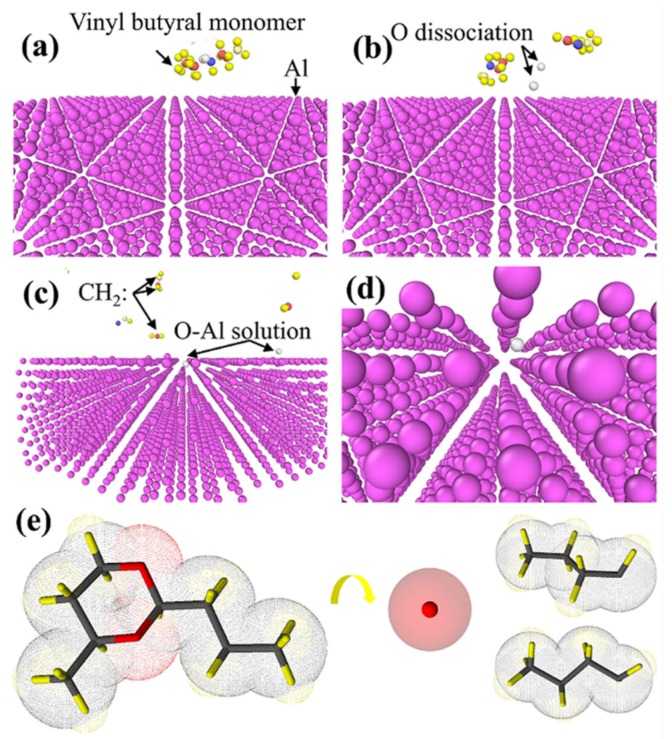
Molecular dynamics simulation of the reaction between vinyl butyral monomer and Al particles: (**a**) initial model of the vinyl butyral monomer and Al particles, (**b**) O atom dissociation process from the vinyl butyral monomer, (**c**) and (**d**) the formation of a solid solution with the Al particles, and (**e**) the actual process of O atom dissociation from the vinyl butyral monomer.

**Figure 3 nanomaterials-10-00438-f003:**
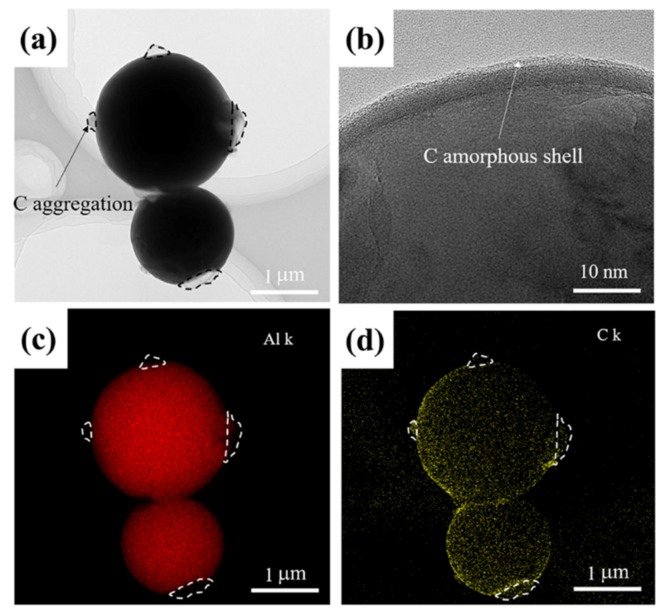
TEM images and EDS maps of the Al particles modified by 3 wt.% PVB: (**a**) TEM image of the Al powder modified by PVB after degradation, (**b**) HRTEM image of the modified Al surface, and (**c**) and (**d**) EDS mapping of the Al and C distribution.

**Figure 4 nanomaterials-10-00438-f004:**
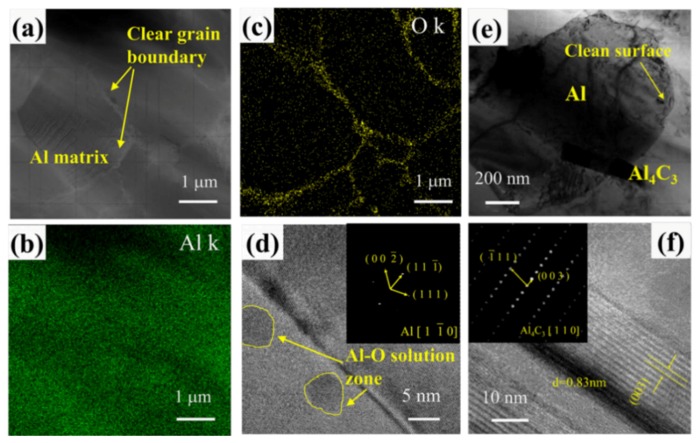
TEM images of the C-O/Al composites modified by 3 wt.% PVB consolidated at 630 °C: (**a**) TEM image of the C-O/Al composite; (**b**) and (**c**) EDS mapping of Al and O distributions, respectively; (**d**) HRTEM of the Al-O microstructure and its selected area electronic diffraction pattern; (**e**) TEM images of the Al_4_C_3_ distribution in the Al matrix; and (**f**) HRTEM of the Al_4_C_3_ microstructure and its selected area electronic diffraction.

**Figure 5 nanomaterials-10-00438-f005:**
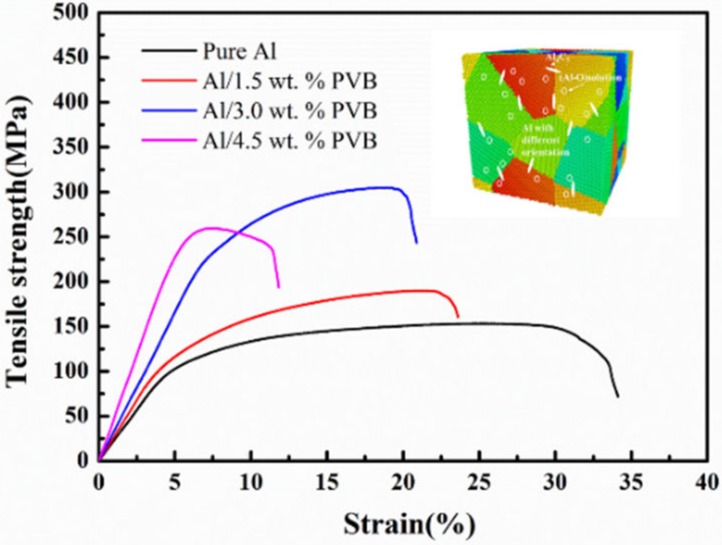
The tensile strength of the C-O/Al composite modified by different PVB contents and pure Al sintered at 630 °C.

**Table 1 nanomaterials-10-00438-t001:** C and O content of the modified powder and calculated quantity of Al-O solution and Al_4_C_3_ in the composites.

Al Modified by PVB Content (wt.%)	Modified Powders		Bulk Composite (Calculation Result)	
	C Content (wt.%)	O Content (wt.%)	Al_4_C_3_ Content (wt.%)	Al-O Solution Content (wt.%)
1.50	0.513	0.019	1.538	0.032
3.00	1.036	0.041	3.107	0.069
4.50	1.631	0.063	4.892	0.106

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
