# Peer review of "Design and Synthesis of C-O Grain Boundary Strengthening of Al Composites"

_nanomaterials, 2020, doi:10.3390/nano10030438_

Round 1
Reviewer 1 Report
Title: Design and Synthesis of C-O Grain Boundary Strengthening of Al Composites
The authors followed in situ method to synthesize a C-O shell on Al powder particles and proposed that C-O grain boundary strengthened the Al composites. Detailed structural and morphological studies of these composites were also carried out and the formation mechanism of this structure was explained by a combination of TEM observations and molecular dynamic simulation results. The work is interesting and well written. I recommend publishing this manuscript in nanomaterials.
Author Response
Thank you for your letter and for the reviewers’ comments concerning our manuscript entitled “Design and Synthesis of C-O Grain Boundary Strengthening of Al Composites”. Those comments are very helpful for revising and improving our paper, as well as the important guiding significance to our researches.
Reviewer 2 Report
The manuscript deals with preparation of strenghtened Al-based composites via powder metallurgy.
The Introduction is thorough, however, the authors could also mention the possibility to increase the mechanical properties of powder-based Al-Al2O3 based composites using methods of severe plastic deformation (e.g. Lenka Kunčická, Terry C. Lowe, Casey F. Davis, Radim Kocich, Martin Pohludka,
Synthesis of an Al/Al2O3 composite by severe plastic deformation, Materials Science and Engineering: A, 646, 2015, 234-241.).
The experimental part is well elaborated and the calculations are relevant. Nevertheless, considering the TEM images chains of precipitated particles at the boudaries of Al grains in which can be seen, showing SEM image of the fracture would be interesting ... could the authors add an image of the fractured surface of the composite?
Reviewer 3 Report
Title: Design and Synthesis of C-O Grain Boundary Strengthening of Al Composites
Article Type: Full length article
Manuscript Number: nanomaterials-702150-peer-review-v1
This work presents a method to synthesize a C-O shell on Al powder particles in a vertical tube furnace. The C-O reinforced Al matrix composites were fabricated by a new powder metallurgy (PM) method associated with the hot pressing technique. The formation mechanism of this structure was explained by a combination of TEM observations and molecular dynamic simulation results. These results indicate the excellent properties of the C-O/Al-strengthened structure.
My recommendation is that the authors carefully consider the below points, revise appropriately.
1.The authors should consider some representative word in the keywords.
2. The authors should check the format of authors in this article. (use “,” or “and”?)
3. Could the authors explain the reason why the PVB additive using 3%? Whether the YS as well as UTS positive correlation with the quantities of PVB additive or not?
4.More different PVB recruitments are encouraged and the results may get more constructive mechanical properties.
5.My suggestion is that the nanoparticles distribution of Al4C3 in the grain as well as the Al-O-Al distribution between aluminium grains should be discussed quantitatively.
Round 2
Reviewer 3 Report
Title: Design and Synthesis of C-O Grain Boundary Strengthening of Al Composites
Article Type: Full length article
Manuscript Number: nanomaterials-702150-peer-review-v2
This work presents a method to synthesize a C-O shell on Al powder particles in a vertical tube furnace. The C-O reinforced Al matrix composites were fabricated by a new powder metallurgy (PM) method associated with the hot pressing technique. The formation mechanism of this structure was explained by a combination of TEM observations and molecular dynamic simulation results. These results indicate the excellent properties of the C-O/Al-strengthened structure.
The rewritten manuscript can be considered to publish after carefully check the English sentence and symbol in full article. Especially, the nomenclature of chemicals or symbol must follow the IUPAC. The authors should carefully check the name and symbol in full article. Such as line 109~110 at Fig. 2(a), the authors should use the PVB instead of PVB monomer for the PVB not a monomer.
